# Wound Repair of the Cell Membrane: Lessons from *Dictyostelium* Cells

**DOI:** 10.3390/cells13040341

**Published:** 2024-02-14

**Authors:** Shigehiko Yumura

**Affiliations:** Graduate School of Sciences and Technology for Innovation, Yamaguchi University, Yamaguchi 753-8511, Japan; yumura@yamaguchi-u.ac.jp; Tel.: +81-83-933-5769

**Keywords:** actin, cell membrane, calcium ion, membrane-remodeling, myosin, wound repair

## Abstract

The cell membrane is frequently subjected to damage, either through physical or chemical means. The swift restoration of the cell membrane’s integrity is crucial to prevent the leakage of intracellular materials and the uncontrolled influx of extracellular ions. Consequently, wound repair plays a vital role in cell survival, akin to the importance of DNA repair. The mechanisms involved in wound repair encompass a series of events, including ion influx, membrane patch formation, endocytosis, exocytosis, recruitment of the actin cytoskeleton, and the elimination of damaged membrane sections. Despite the absence of a universally accepted general model, diverse molecular models have been proposed for wound repair in different organisms. Traditional wound methods not only damage the cell membrane but also impact intracellular structures, including the underlying cortical actin networks, microtubules, and organelles. In contrast, the more recent improved laserporation selectively targets the cell membrane. Studies on *Dictyostelium* cells utilizing this method have introduced a novel perspective on the wound repair mechanism. This review commences by detailing methods for inducing wounds and subsequently reviews recent developments in the field.

## 1. Introduction

The cell membrane serves as a crucial barrier between the extracellular and intracellular spaces, yet it is consistently vulnerable to physical or chemical damage. Such injuries compromise the membrane’s integrity, leading to an influx of undesirable substances into the cell and cytoplasmic loss. Local wounds on the cell membrane also impact cell polarity during migration [1] and influence the division axis and symmetrical division in cell division [2]. Mechanically active tissues, like mammalian skeletal and cardiac muscles, frequently experience membrane wounds due to repeated contractions [3,4,5]. Ischemia-reperfusion injury followed by heart attack and stroke also damages the cell membrane [6]. Infection by pathogenic funguses, bacteria, and viruses and their pore-forming toxins can also result in membrane wounds at the cell membrane [7,8,9,10]. Loss of the wound repair function is observed in various diseases, including diabetes [11], muscular dystrophies [12,13], acute kidney injury [14], and vitamin deficiencies [15]. Recent studies have identified defects in wound repair as common in Parkinson’s and Alzheimer’s diseases [16,17,18]. Plant cells, affected by freezing damage in cold seasons, also possess the capability to repair damaged membranes [19,20,21]. Similar to DNA repair, wound repair is a physiologically vital phenomenon for living cells. Moreover, many methods for introducing extracellular substances into cells, such as microinjection and electroporation, rely on cellular wound repair. See also recent good reviews [6,22,23,24,25,26].

The mechanisms of wound repair have been extensively studied across various model organisms, including mammalian cells [27,28], amphibian oocytes [29,30,31,32], echinoderm oocytes [33,34,35], fruit flies [36,37,38], nematodes [12,39,40], amoebae [41,42], yeast [43,44], ciliate [45], plant cells [19,46,47], and *Dictyostelium* cells [48]. A common feature among these mechanisms is the essential role of Ca^2+^ influx from an external medium in the wound repair process. While the “membrane patch hypothesis” suggests that cytosolic membrane vesicles accumulate at the wound site to form an impermanent “patch” for emergency wound pore plugging [34,49,50,51], alternative hypotheses that do not involve patching have also been proposed [25,27,52]. However, there is no universally accepted general model for the mechanisms driving the repair process.

In larger cells like *Xenopus* oocytes and *Drosophila* embryos, an actomyosin ring surrounds the wound site, similar to the contractile ring in dividing cells, facilitating wound pore closure [36,53,54,55]. Conversely, in smaller cells like yeast, animal culture cells, and *Dictyostelium* cells, actin transiently accumulates at the wound site [43,56,57,58]. The absence of actin polymerization hinders wound pore closure in, mammalian culture cells, muscle cells, and *Dictyostelium* cells [51,59,60]. Notably, myosin II’s accumulation at the wound site and its contribution to wound repair in small cells remain controversial [43,57,58,61].

This article reviews recent advancements in cellular wound repair, with a specific focus on the wound response and repair process in *Dictyostelium* cells. The discussion encompasses techniques for studying the wound repair mechanism, an overview of previous information on wound repair, and future perspectives in the field.

## 2. The Cell Can Repair a Wounded Cell Membrane

The presence of a cell membrane wound repair mechanism must be noticed during the initial phases of single-cell microsurgery and microinjection experiments [30,31,62,63,64,65]. For example, when a *Dictyostelium* cell is divided into two fragments using a microneedle, the nucleate fragment exhibits normal migration, whereas the anucleate fragment is incapable of doing so [66]. This experiment underscores the nucleus’s indispensable role in cell migration and simultaneously emphasizes the prompt repair of the cell membrane, which was wounded during microsurgery—a pioneering observation in *Dictyostelium* cells.

The majority of wound experiments have focused on a limited life stage of cells. The life cycle of *Dictyostelium discoideum* is broadly categorized into four stages: vegetative, aggregation, multicellular, and culmination. After the starvation of vegetative cells, individual cells aggregate to form streams towards the aggregation center. Aggregation is mediated by the chemotaxis of cells toward cAMP excreted from the aggregation centers. This process results in the formation of a multicellular organism and eventually leads to the development of fruiting bodies consisting of spores and stalks. Wound repair is observed at all stages in *Dictyostelium* cells (Figure 1A), including spore cells, which are dormant cells with a rigid cell wall. Furthermore, wound repair is noted at different stages of the cell cycle, such as interphase and the mitotic stage, in *Dictyostelium* cells [2].

## 3. Monitoring of Wound Repair

Various methods have been employed to investigate the wound repair mechanism. In early experiments, the cell membrane was wounded mainly by microneedle poking in large cells such as protozoan amebae [41,65,69], amphibian eggs [30,62,64], and echinoderm eggs [50]. For small cells like yeast, animal cultured cells, and *Dictyostelium* cells, laser ablation has been predominantly used due to the technical challenges and time constraints associated with microneedle poking in such small cells. Recent research also utilizes laser ablation for large cells, offering precise-sized wounds and accurate timing. However, both laser ablation and previous methods not only damage the cell membrane but also impact intracellular structures, including cortical actin networks, microtubules, and organelles.

To address this, we have developed an improved laser ablation method that selectively injures only the cell membrane. As depicted in Figure 1B, after placing cells on a carbon or gold-coated coverslip, a laser beam is focused on the coat underneath the cells. The laser energy absorbed by the coat generates heat and/or plasmon [48,70], selectively injuring the cell membrane attached to the coat [48]. This method has been originally invented for the introduction of extracellular substances into cells [71]. Instead of wounding individual cells, for biochemical analysis, a large number of cells can be wounded by treating with pore-forming agents or detergents [72,73,74].

For monitoring the wounding process, propidium iodide (PI) or FM1-43 has been widely used. PI, a cell-impermeant dye emitting fluorescence upon binding to RNA or DNA, and FM1-43, a cell-impermeable fluorescent lipid analog emitting fluorescence upon insertion into the membrane, are placed in the external medium. Their entry into the cytosol is monitored by the increase in fluorescence upon wounding. As shown in Figure 1C, PI fluorescence begins to increase at the wound site upon injury, spreading over the cytosol, suggesting PI entry through the wound pores. Figure 1D (BSS) illustrates the time course of PI fluorescence intensity in the cytosol of wounded cells, indicating that PI influx ceases within 2–3 s after injury, terminating urgent wound repair within this timeframe. Figure 1E demonstrates the influx of FM1-43 dye upon wounding, also showing that the dye enters from the wound pore and spreads across the cytoplasm.

To visualize the wound pore in *Dictyostelium* cells, cells expressing GFP-cAR1 (cAMP receptor) as a membrane protein marker are wounded. Immediately after wounding, a black spot appears at the laser application site (Figure 1F). This black spot is not generated by photobleaching, as it transiently expands slightly, then shrinks, and eventually closes (Figure 1G). This closure is not uniform but occurs from the wound edge to the center.

## 4. Ca^2+^ Influx as the First Signal

The initial signal common to all examined cells across various species is the influx of Ca^2+^ from the wound pore [30,32,41,50,75]. Monitoring this influx is feasible using a Ca^2+^ indicating fluorescent dye or a GFP-based Ca^2+^ indicator. Figure 1H presents a time series of fluorescence images of *Dictyostelium* cells expressing GCAMP6s, a GFP-based fluorescent Ca^2+^ indicator. In Figure 1I (BSS), the time course of fluorescence intensities in the cytosol is depicted. Intracellular Ca^2+^ concentration (Ca_i_^2+^) promptly rises upon wounding, returning to resting levels within approximately 7 s. In the absence of external Ca^2+^, Ca_i_^2+^ remains unchanged upon wounding (Figure 1I, EGTA), indicating that the influx of Ca^2+^ triggers the increase in Ca_i_^2+^. Without external Ca^2+^, PI influx persists, leading to the eventual death of wounded cells (Figure 1D, EGTA, and Figure 1J). Additionally, the black spot observed in experiments using GFP-cAR1 does not close without Ca^2+^ influx. A concentration higher than 0.1 mM of Ca^2+^ in the external medium is necessary for wound repair in *Dictyostelium* cells [51].

Ca^2+^ influx induces the release of Ca^2+^ from intracellular stores through the calcium-induced calcium release (CICR) mechanism [76,77]. Deleting CICR reduces the amplitude of Ca_i_^2+^ but does not impact wound repair, indicating that a local increase in Ca_i_^2+^ is crucial, not a global one [67]. On the other hand, MCOLN1, an endosomal and lysosomal Ca^2+^-channel, is crucial for cell membrane repair in muscle cells, emphasizing the significance of Ca^2+^ release from intracellular stores in wound repair [78].

Various intracellular targets of Ca^2+^ for wound repair include dysfelin, mitsugmin 53 (MG53), neuroblast differentiation-associated protein (AHNAK), calpain, calmodulin, annexins, the endosomal sorting complex required for transport (ESCRT), protein kinase C, and actin-related proteins. These will be discussed in detail later.

## 5. Closing Wound Pores

### 5.1. Spontaneous Self-Sealing

Ruptured artificial lipid bilayer membranes exhibit spontaneous resealing [79]. Similarly, extremely small wound pores, such as those generated by electroporation in live cells at the nanometer scale, are thought to undergo spontaneous resealing due to thermodynamically unfavorable lipid disorder (Figure 2A). However, even with such small pores, there may be a necessity for an active wound repair mechanism [80]. Additionally, membrane pores created by pore-forming toxins, despite being nanometer-sized, require an active wound repair mechanism [7,23,81].

### 5.2. Self-Sealing by Regulation of Surface Tension

Given the tension on the cell surface, larger pores cannot spontaneously reseal against the cell surface tension. The wound-induced influx of Ca^2+^ triggers the fusion of exocytic vesicles with the cell membrane, extending beyond the wound site and enlarging the plasma membrane (Figure 2B). This process results in a reduction of cell surface tension, facilitating spontaneous resealing and closure of the wounded pore [82]. In large cells, although the actomyosin ring exerts force to close the wound pore against the opening force of the cell surface tension, self-sealing alone is insufficient, and a membrane patch is also required for sealing, as described later.

### 5.3. Sealing by Protein Aggregation

The wounded pores have been suggested to be clogged by the aggregation of proteins, including annexins and actin (Figure 2C). Annexin A5 self-assembles into two-dimensional arrays on the membrane upon Ca^2+^ activation, a crucial aspect of its role in plasma membrane repair in mammalian cells [83]. Similar clogging phenomena have been reported for other annexins, which collaborate with wound repair-related proteins such as actin, dysferin, and MG53 [84,85,86,87]. We will delve into these proteins in more detail later. Actin accumulates at the wound site, potentially serving a clogging function through actin gelation. However, this accumulation does not happen immediately upon wounding but occurs after the cessation of PI influx.

### 5.4. Sealing by Membrane Patch

Mammalian red blood cells, lacking endomembranes, including nuclei, take a longer time to reseal wound membranes or fail to repair in physiological conditions, suggesting that endomembranes are necessary for wound repair [88]. The membrane patch hypothesis was initially proposed in large cells like echinoderm and frog oocytes (Figure 2D). A local increase in Ca^2+^ induces the fusion of small cytoplasmic vesicles with each other, creating a continuous membrane plug at the wound site along with the plasma membrane [34,35]. More recently, cortical granules in Xenopus oocytes have been identified as such intracellular compartments, and their fusion was visualized in live cells [89,90]. This wound repair process is succeeded by the constriction of an actomyosin ring, akin to the contractile ring involved in cytokinesis.

Various sources for the membrane patch, including lysosomes [91,92], endosomes [93,94], MG53-rich vesicles [75], dysferlin-containing vesicles [95,96], or AHNAK-positive “enlargeosomes” [97,98], have been proposed. However, these vesicles and organelles might not meet the spatiotemporal requirements for rapid and efficient wound repair, considering the possibility of multiple wounds with very short intervals [67].

Recently, we proposed that the vesicles for the membrane plug are newly generated at the wound site in *Dictyostelium* cells [51]. In influx experiments using FM dye, most of the FM fluorescence diffuses in the cytosol, but a portion of FM dye remains at the wounded site and increases in size (Figure 1E), indicating membrane accumulation at the wound site. In the PI influx experiment (Figure 1C), a portion of PI fluorescence also persists at the wound site, suggesting that cytoplasm, including PI dye, is entrapped in the newly enclosed vesicles. It is improbable that the membrane plug originates from the broken cell membrane due to the limited amount of the broken cell membrane. Additionally, vesicles are unlikely to be transported from other locations since pharmacological disruption of microtubules and actin did not impede membrane accumulation. Therefore, we propose that the vesicles for the membrane plug are generated de novo at the wound site, although the mechanism for this generation remains unclear.

As described earlier, experiments using GFP-cAR1 show that the wound pore is not repaired from the wound edge to the center. Therefore, the wound edge grows toward the center through vesicles repeatedly fusing with the edge of the cell membrane, rather than forming a fused large patch to plug the wound pore.

### 5.5. Endocytosis of Damaged Membrane

Ca^2+^-triggered endocytosis is suggested to eliminate damaged membrane (Figure 2E). The membrane, including the damaged portion, invaginates inward, and the resulting bud is removed by releasing it into the cell, dependent on Ca^2+^ [74,99]. Upon wounding, acid sphingomyelinase is secreted into the extracellular space through Ca^2+^-dependent lysosomal exocytosis. This enzyme hydrolyzes sphingomyelin in the cell membrane into ceramide, facilitating membrane invagination and vesiculation [100]. Ceramide formation by sphingomyelinase also induces caveolae-mediated endocytosis, internalizing the wounded membrane [99,101,102,103,104,105]. It has been also reported that clathrin- and dynamin-mediated endocytosis facilitates removing the wounds by pore-forming proteins or toxins [106,107].

In *Dictyostelium* cells, neither endocytosis nor exocytosis appears to contribute to membrane accumulation for wound repair [51], despite the rapid turnover of the cell membrane through endocytosis–exocytosis coupling [108,109,110,111]. Notably, caveolin proteins are not present in *Dictyostelium* [112] and inhibitors of sphingomyelinase do not affect the wound repair in *Dictyostelium* cells (our preliminary observations). Additionally, clathrin- and dynamin-mediated endocytosis does not contribute to wound repair in *Dictyostelium* cells [48].

### 5.6. Vesicle Budding and Shedding to the Outside

Rather than endocytosis involving the inward budding and scission of the damaged membrane, vesicle budding or blebbing toward the outside of the cell, followed by scission, facilitates the removal and shedding of damaged membrane or pore-forming reagents (Figure 2F). Membrane-binding proteins for wound repair, such as the endosomal sorting complex required for transport (ESCRT) and annexins, facilitate this type of shedding in a Ca^2+^-dependent manner in mammal cells [10,44,84,99,113,114].

In *Dictyostelium* cells, FM dye that accumulates at the wound site remains there for the duration of our observation. The wounded sites do not move relative to the substrate and eventually shed onto the substrate as cells migrate [1].

## 6. Membrane-Binding Proteins in Wound Repair

For wound repair, various membrane-binding proteins, including annexins, the ESCRT complex, synaptotagmin, and dysferin, have been proposed. These proteins also act as sensors, detecting damage to the cell membrane due to their calcium-dependency [115].

### 6.1. Annexins

Annexins, a highly conserved and ubiquitous family of Ca^2+^- and phospholipid-binding proteins, play a crucial role in wound repair [116,117,118,119,120,121]. In vertebrates, 12 annexin subfamilies (A1–A11 and A13) have been identified. Annexins such as A1, A2, A5, and A6 accumulate at the wound site by binding to the inner cell membrane, particularly acidic phospholipids like phosphatidylserine, in response to a Ca^2+^ influx. Through membrane binding and interactions with other proteins, such as S100 family proteins, annexin prevents further expansion of the wound pore, reduces membrane tension, and prepares the membrane for resealing [59,122,123].

Annexins induce curvature in the free-edge membranes and generate constriction force to close the wound pore through annexin crosslinking [124,125,126]. Annexins can also be cross-linked by transglutamilases in a Ca^2+^-dependent manner [127], which have also been implicated in plasma membrane repair [128].

Moreover, annexins have been proposed to assemble into multimeric lattice structures, recruiting M53-laden vesicles and mini-dysferlin72, effectively clogging the wound pore [13,83,120,129]. Some annexins, such as annexin 1 and 2, can bind to actin and stabilize actin filaments at the wound site [85,119,130,131,132].

*Dictyostelium* possesses two annexin genes, annexin C1 (annexin VII or synexin) and annexin C2 (annexin I) [133]. Both can bind phosphatidylserine in a Ca^2+^-dependent manner [134,135]. Only annexin C1 accumulates at the wound site immediately after wounding. Wounded annexin C1-null cells exhibit irregular curves with multiple peaks in PI influx, Ca^2+^ influx, and actin dynamics. Additionally, annexin C1-null cells have a significantly reduced survival rate following injury, suggesting that annexin C1 partially contributes to wound repair in *Dictyostelium* cells [48,67].

### 6.2. ESCRT Complexes

The endosomal sorting complex required for transport (ESCRT) is categorized into five protein complexes (ESCRT-0, ESCRT-I, ESCRT-II, ESCRT-III, and Vps4). These complexes play integral roles in various cellular processes, including endosomal budding transport, virus budding, and cytokinesis. The ESCRT complex constricts and severs narrow necks during membrane budding processes [44,136,137,138,139,140,141,142,143,144]. Additionally, ESCRT complexes have been implicated in shedding wounded membranes as extracellular vesicles [44,114,144]. Upon injury, ESCRT complexes promptly accumulate at the wound site, protrude the wounded membrane as a bud or bleb, and subsequently cut it off to release extracellular vesicles. This ESCRT-mediated abscission of the wounded membrane appears to limit smaller-sized wounds (<100 nm in diameter) [145,146].

Interestingly, ESCRT complexes also participate in repairing damaged membranes of intracellular organelles, such as the nuclear envelope and lysosomes [147,148,149]. Furthermore, ESCRT complexes mediate the sealing of holes in the nascent nuclear envelope and nascent autophagosome [150].

While ESCRT complexes themselves are not sensitive to Ca^2+^, Ca^2+^-sensitive proteins like ALG-2 and calmodulin confer Ca^2+^ sensitivity on ESCRT complexes [10,144,151]. Recently, it has been reported that annexin A6 also plays a similar role in the secretion of exosomes [152].

In *Dictyostelium* cells, components of the ESCRT complexes accumulate at the wound site immediately upon injury, depending on the influx of Ca^2+^ [51]. However, in ESCRT null cells, PI influx ceases normally, and actin dynamics are observed, similar to wild-type cells. This suggests that ESCRT complexes are not essential for wound repair in *Dictyostelium* cells [48].

During cytokinesis in animal cells, ESCRT complexes and annexins accumulate at the cleavage furrow and/or midbody and are considered to play a role in cytokinesis [153,154,155,156,157,158]. *Dictyostelium* cells lack a midbody and undergo division through physical cutting via the constriction of the contractile ring and the traction force of the two daughter fragments migrating in opposite directions [159,160]. This suggests that this abscission might be a form of ‘physiological wound’. However, our preliminary observations indicate that neither ESCRT components nor annexins accumulate at the torn edges, suggesting the existence of a novel mechanism for cytokinetic abscission in *Dictyostelium* cells.

### 6.3. Synaptotagmin

Synaptotagmin comprises a family of Ca^2+^-binding and membrane-trafficking proteins, with particular emphasis on its well-characterized role in the release of synaptic vesicles in neurons, where it regulates Ca^2+^-dependent exocytosis [161]. Synaptotagmin 7, specifically, participates in the repair of wounded membranes through Ca^2+^-dependent lysosome exocytosis [92,162]. Knockdown mice lacking synaptotagmin 7 exhibit defects in wound repair [163]. While *Dictyostelium* possesses a synaptotagmin-like protein, there is currently no available information regarding its role in wound repair.

### 6.4. Dysferlin

Dysferlin, a membrane protein within the Ferlin family involved in vesicle fusion, is notably abundant in skeletal and cardiac muscle [164]. Dysferlin binds to vesicles containing acid phospholipids, such as phosphatidylserine, through the dysferlin C2 domain, relying on Ca^2+^ [165]. Mice deficient in dysferlin display defects in wound repair in muscle cells and develop muscular dystrophy [12]. Dysferlin is proposed to generate a membrane patch by recruiting and fusing vesicles in wounded skeletal and cardiac muscle cells [96,166]. Dysferlin organizes vesicle fusion with the assistance of binding partners like S100A10, annexin A2, AHNAK, caveolin-3, and TRIM72 (MG53) [13,75,102,115,167]. TRIM72, also known as MG53, which is highly expressed in muscle cells, assembles into a higher-ordered structure on the phosphatidylserine-enriched membranes. This assembly and association with the membrane depend on its oligomeric assembly and ubiquitination activity, facilitating vesicle transport to the wound site [168]. Notably, *Dictyostelium* cells lack a dysferlin homolog.

## 7. Cytoskeletons

The most comprehensively understood molecular mechanism for wound repair is derived from research in *Xenopus* oocytes. The entry of Ca^2+^ activates the small G-protein Rho, leading to the accumulation of myosin II [53,55]. Myosin II is a type II myosin that can assemble into bipolar filaments, generating a contractile force by interacting with actin filaments. Ca^2+^ influx also activates cdc42, another small G-protein, leading to actin assembly [90,169,170,171,172]. An actomyosin ring structure, akin to the contractile ring during cytokinesis, forms around the wound pore, facilitating wound closure. However, the appearance of the actomyosin ring is limited to large cells. In this section, we will explore the contributions of actin, myosin, actin-binding proteins, and microtubules.

### 7.1. Actin

Actin transiently accumulates at the wound site upon injury in various organisms [173]. Previous studies on cultured mammalian cells have indicated that, before wound-induced actin accumulation, pre-existing cortical actin networks at and around wound pores are largely removed [122,174,175]. This removal is believed to be necessary for the access of exocytic vesicles to the cell membrane and subsequent actin accumulation at the wound site [91,176]. Although the molecular mechanism of actin removal is not fully understood, Ca^2+^-dependent proteases such as calpain [177,178,179,180] or Ca^2+^-dependent actin-depolymerizing factors, such as actin-related proteins severing or depolymerizing actin filaments, might remove the actin cortex.

Conversely, in *Dictyostelium* cells, the removal of cortical actin networks is substantially undetectable upon wounding [48]. This discrepancy may be caused by differences in wounding methods. Conventional wounding methods disrupt not only the cell membrane but also the actin cortex, microtubules, and organelles, whereas the method used for *Dictyostelium* cells only disrupts the cell membrane, not the inner structures. Disruption of Ca^2+^-storing organelles such as the endoplasmic reticulum might result in the disassembly of actin filaments by activating Ca^2+^-dependent proteases or Ca^2+^-dependent actin-depolymerizing factors. Alternatively, although the dynamic instability of actin filaments is shown only in vitro, in contrast to that of microtubules [181,182], disassembly of actin filaments might expand to a much larger area due to catastrophic disassembly when a part of the meshwork is disrupted by the conventional wounding method.

Figure 3A–C show typical time courses of fluorescence images and fluorescence intensities of GFP-lifeact, a marker of actin filaments expressed in *Dictyostelium* cells upon wounding. Actin transiently accumulates at the wound site. In the presence of latrunculin A, a depolymerizer of actin filaments, actin does not accumulate at the wound site, and PI influx does not stop (Figure 3D, LatA), resulting in the failure of wound repair [51]. Therefore, actin accumulation is essential for wound repair.

There are two possible mechanisms for actin accumulation at the wound site: (1) preexisting cortical actin filaments flow (moving along the cell membrane) toward the wound site (flow model), and (2) monomeric actin polymerizes at the wound site (de novo synthesis model). In *Xenopus* oocytes, the actomyosin ring is generated by the flow of cortical actin filaments toward the wound site, accompanied by dynamic actin polymerization [183]. However, such a flow is not observed around the wound site in *Dictyostelium* cells with curable wounds [48].

When cells are incubated with jasplakinolide, a stabilizer for actin filaments, for 30 min and then with latrunculin A, the number of polymerizable actin monomers should be significantly reduced, although the pre-built actin structures are stabilized by jasplakinolide. Upon wounding, actin does not accumulate at the wound site (Figure 3E, Jasp + LatA), suggesting that the de novo synthesis model is preferable because actin cannot newly polymerize without polymerizable actin monomers in the presence of both inhibitors. Therefore, it is plausible that actin polymerizes de novo at the wound site.

Although the aggregation of actin filaments itself may directly plug the wound pore, this is not the case in *Dictyostelium* cells because the wound pore substantially closes about 2 s after wounding, as observed from the influx of PI. On the other hand, actin begins to accumulate about 2.5 s after wounding. As shown in Figure 3D, PI influx in the presence of latrunculin A shows two phases: an initial rapid influx phase and a subsequent slow influx phase. PI influx in the control substantially stops after the initial phase. Therefore, the role of actin is not to directly constrict or plug the pore but rather to maintain pore closure after the emergent closing by the membrane plug.

Figure 3F shows the time course of the fluorescence intensity of FM1-43 at the wound site in the presence and absence of latrunculin A. The membrane accumulates at the wound site by two steps in the control (BSS): the initial rapid accumulation upon wounding (urgent membrane plug) and thereafter gradually increasing accumulation at a very low level. In the presence of latrunculin A, the size of the membrane pore gradually increases, and an increasing amount of membrane accumulates, suggesting an additional role for actin accumulation to prevent further membrane accumulation, although the detailed mechanism is elusive. Since PI influx does not cease in the presence of latrunculin A, the urgent membrane plug in the first step is incomplete, and the actin accumulation plays a role in the completion of the plug.

### 7.2. Myosin II

To explain the actin ring constriction mechanism independent of myosin II, it is proposed that actin continuously assembles on the inside of the actin ring and disassembles at the outer edge (actin treadmilling mechanism) in *Xenopus* oocytes [52].

Myosin II does not accumulate at the wound site in small cells such as yeast, *Dictyostelium*, and cultured animal cells. Moreover, pharmacological inhibition has demonstrated the dispensability of myosin II in *C. elegans* hypodermal cells [184] and sea urchin coelomocytes [185]. Conversely, in mammalian cultured cells, myosin IIA interacts with MG53 to regulate vesicle trafficking to the wound site [186], while myosin IIB facilitates wound-induced exocytosis and the membrane resealing process [57].

*Dictyostelium* cells possess a single copy of the myosin II (heavy chain) gene. Interestingly, myosin II disappears from a much larger area than the wound size (Figure 3G). Furthermore, myosin II null *Dictyostelium* cells can repair wounds comparably to wild-type cells [68]. Consequently, myosin II is not essential for wound repair in *Dictyostelium* cells. Notably, the disappearance of myosin II filaments is independent of both its phosphorylation and motor activities, suggesting that it occurs without accompanying the disassembly of myosin II filaments [68,187,188]. A local decrease in tension in the cortical actin network at the wound region may release myosin II filaments from the cortex, as myosin II filaments preferentially bind to stretched actin filaments [189].

One plausible reason why small cells do not rely on myosin II is that the power of myosin II may be necessary to prevent the wound pore from opening in large animal cells due to the significantly higher surface tension in large cells compared to smaller cells. Another reason may involve a different type of myosin replacing myosin II to contribute to the closing constriction power of the wound pore. Groups of type I myosins, MyoB and MyoC, transiently accumulate at the wound site in *Dictyostelium* cells [48]. A third reason is the notable difference in wound sizes used for the experiments between large and small cells: about 100 µm in diameter for the former and around 1 µm.

When *Dictyostelium* cells are wounded with different sizes, wounds larger than 1.5 µm in diameter result in cell lysis with a 50% frequency. Most cells subjected to 2 µm wounds cannot survive [67]. However, in such significantly wounded cells, we observed a ring composed not only of actin but also myosin II filaments (unpublished data). If the wound size is 1.5 µm in diameter, the wound area is estimated to occupy 1.2% of the whole cell surface area for *Dictyostelium* cells. On the other hand, for *Xenopus* oocytes, if the wound size is 100 µm in diameter, it is estimated to be 8%, suggesting that *Xenopus* oocytes can tolerate much larger wounds than *Dictyostelium* cells. Interestingly, actin and myosin II flow toward the wound center in such significantly wounded *Dictyostelium* cells, resembling the flows observed in *Xenopus* oocytes (unpublished data). The participation of myosin II may depend on the wound size and the cell surface tension.

### 7.3. Actin-Related Proteins (ARPs)

The transient accumulation of actin at the wound site is orchestrated by the assembly and subsequent disassembly of actin filaments, a process regulated by actin-related proteins (ARPs). Various direct modulators of actin polymerization, such as formins, Arp2/3 complex, SCAR/WAVE (suppressor of cAR/WASP family verprolin homologous protein), and WASP (Wiskott-Aldrich Syndrome protein), have been identified as participants in actin assembly at the wound site across multiple organisms [175,185,190,191,192]. Formins and Arp2/3 complex direct elongation of unbranched and branched actin filaments, respectively. Arp2/3 complex is activated by WASP family proteins such as WASP, SCAR/WAVE, whereas formins are individually active.

In *Dictyostelium* cells, 13 out of 27 examined ARPs accumulate at the wound site. Severin (an F-actin-severing protein akin to gelsolin) rapidly accumulates at the wound site, followed by WASP, Arp3 (a component of the Arp2/3 complex activated by WASP), ABP34 (an actin-bundling protein), and MyoB (a type I myosin), which accumulate significantly earlier than actin. Alpha-actinin (an actin-crosslinking protein), filamin (another actin-crosslinking protein), and CAP32 (an actin capping protein) begin to accumulate simultaneously with actin. MyoC (a type I myosin), CARMIL (a multidomain scaffold protein), and fimbrin (an actin-bundling protein) start accumulating after actin. Coronin and cofilin accumulate during the actin disassembly stage, consistent with the consensus that coronin inactivates the Arp2/3 complex [193], and ADF/cofilin promotes actin disassembly [194].

Similar to myosin II, EfaA1 (elongation factor 1 alpha 1, an actin-bundling protein), DlpA (dynamin-like protein A), MHCKC (myosin heavy chain kinase C), and cortexillins A and B (actin-binding proteins) transiently disappear from the wound site [68]. Intriguingly, with the exception of EfaA1, most of these proteins localize at the cleavage furrow during cytokinesis and at the rear regions during cell migration [195,196,197,198,199,200]. Incidentally, signaling proteins localizing at the cleavage furrow, such as PakA (p21-activated protein kinase A) [201], PTEN (phosphatase and tensin homolog deleted on chromosome 10) [202,203], and GAPA (IQGAP-related protein) [204], also transiently disappear at the wound site. Figure 4 provides a summary of the durations of their appearance and disappearance.

The necessity of these ARPs for wound-induced actin dynamics has been explored through inhibition experiments using knockout mutants or pharmacological inhibition in *Dictyostelium* cells. Inhibition of WASP, Arp2/3, formin, and profilin significantly reduces actin accumulation at the wound site [48]. Although most ARPs accumulating at the wound site are found to be dispensable, they likely possess redundant roles in the regulation of the actin cytoskeleton, a hypothesis that warrants further investigation using double or triple knockout mutants [205].

### 7.4. Microtubules

Microtubules exhibit distinct behaviors in wound repair across different organisms. In *Xenopus* oocytes, microtubules accumulate and form a radial array around the wound site [191]. In epithelial kidney cells, microtubules locally disassemble upon wounding and elongate toward the wound site [206]. However, in *Drosophila* embryos, microtubules do not undergo significant changes upon wounding, yet pharmacological disruption of microtubules reduces wound repair [36]. Microtubules have been reported to transport small vesicles, such as lysosomes and vesicles derived from the Golgi apparatus, to the wound site, relying on motor proteins like kinesins and myosins [96,207,208]. In addition, it has been reported that microtubules regulate the actomyosin recruitment to the wound site in *Xenopus* oocytes [191].

In contrast, microtubules neither accumulate nor elongate toward the wound site upon wounding in *Dictyostelium* cells. Wound repair defects are not detected in the presence of a microtubule depolymerizer [51]. Therefore, vesicles are not transported along microtubules, and microtubules are not essential for wound repair in *Dictyostelium* cells. In these cells, the membrane at the wound site is not transported from other locations; instead, vesicles are generated de novo at the wound site, as mentioned earlier.

## 8. Signals for the Wound Repair

The downstream signals following Ca^2+^ influx upon wounding have been investigated. The Ca^2+^-sensitive membrane-binding proteins as described above (Section 6) can directly play a role in wound repair upon influx of Ca^2+^. We will see other signal pathways.

### 8.1. Protein Kinase C

Protein kinase C (PKC), activated by signals such as increases in Ca_i_^2+^, is implicated in actin assembly in *Xenopus* oocytes and yeast during wound repair [43,209]. Elevating Ca_i_^2+^ levels through treatment with a Ca ionophore results in the transient translocation of actin to the cell cortex [210,211,212]. PKC has been proposed to regulate small G proteins, which in turn regulates actin-related proteins for dynamics of actin at the wound site [170,213]. However, PKC does not seem to contribute to the wound repair in *Dictyostelium* cells [48].

### 8.2. Small G Proteins

The Rho family of small G proteins (Rho, Rac, Cdc42) serves as an upstream signal for regulating actin dynamics at the wound sites in *Xenopus* and *Drosophila* oocytes. These GTPases, including Rho, Rac, and Cdc42, are recruited to wounds in distinct spatiotemporal patterns [37,171,214]. Through their downstream effectors such as WASP, Scar/WAVE, and WASH, they recruit actin to the wound site, forming an actomyosin ring along the wound edge [172,190]. The Rab family of small G proteins regulate vesicle transport, including vesicle attachment to motor proteins and their tethering to target membranes [215]. The Rab3a, one of the Rab family, binds to lysosomes with myosin II heavy chain A, which is required for the lysosome exocytosis for wound repair in HeLa cells and melanocytes [216,217].

*Dictyostelium* cells possess 20 Rac proteins but lack Rho and Cdc42 [218]. A Rac inhibitor significantly diminishes actin accumulation amplitude and delays both the initiation and termination times, indicating that Rac regulates wound-induced actin dynamics. The specific type of Rac contributing to the actin accumulation requires further clarification. The Ras family also regulates actin polymerization, the deletion of both RasG and RasC does not affect wound-induced actin accumulation [48].

### 8.3. Reactive Oxygen Species (ROS)

The wound-induced influx of Ca^2+^ causes mitochondria to associate with the wounded membrane [219] and activates the uptake of Ca^2+^ into mitochondria in skeletal muscle cells, which generates reactive oxygen species (ROS). ROS locally activate RhoA, triggering actin accumulation at the wound site [40,220,221]. It is also reported that oxidation mediates assembly of MG53 for wound repair [75].

### 8.4. Calmodulin

Calmodulin, a multifunctional Ca^2+^-binding messenger protein, is reported to contribute to cellular wound repair in neurons, green algae, and *Dictyostelium* cells [51,222,223]. Calmodulin activates calpain, a Ca^2+^-dependent protease, to degrade fodrin, an actin-related protein, and thereby disassembles the actin cortex, which facilitates the resealing of wounded membrane in neurons and axons [224].

In *Dictyostelium* cells, calmodulin transiently accumulates at the wound site immediately after injury, depending on Ca^2+^ influx. In the presence of W7, a calmodulin inhibitor, calmodulin fails to accumulate at the wound site, and wound-induced PI and FM1-43 influx persists, contrary to the control, indicating the essential role of calmodulin in wound repair. W7 inhibits annexin accumulation but does not affect ESCRT complexes accumulation. Notably, calmodulin may regulate ESCRT complexes for exosomal biogenesis in human cultured cells [225]. W7 inhibits actin accumulation, while latrunculin A does not inhibit calmodulin accumulation in *Dictyostelium* cells, suggesting that calmodulin acts upstream of annexin and actin.

Chemotaxis signaling pathways, including the regulation of dynamics in the membrane and cytoskeleton, have been extensively investigated in *Dictyostelium* cells as well as neutrophils [226,227]. We explored the wound repair signaling pathways by referencing the chemotaxis signaling pathways. Figure 5 provides an overview of the signaling pathways involved in wound repair in *Dictyostelium* cells. After the influx of Ca^2+^ upon wounding, calmodulin and annexin C1 accumulate immediately at the wound site, which triggers the de novo generation of vesicles and mutual fusion of vesicle–vesicle and vesicle–cell membrane to make an urgent membrane plug. The TORC2, Dock/Elmo, PIP2-derived product, and PLA2 pathways are activated, which is common in the chemotaxis signaling pathway. Racs, WASPs, and then formins and Arp2/3 are involved in these pathways, and further downstream, many ARPs regulate the actin dynamics at the wound site.

## 9. Wound Repair Model for *Dictyostelium* Cells

Figure 6 provides a comprehensive overview of wound repair in *Dictyostelium* cells. Upon wounding, the influx of Ca^2+^ through the wound pore serves as a signal for the accumulation of calmodulin and annexin C1 at the wound site, triggering the formation of an urgent membrane plug. This process involves the de novo generation of membrane vesicles, followed by their mutual fusion with both other vesicles and the cell membrane.

In the second step, actin accumulates through de novo polymerization to complete the membrane plug. The signals for actin accumulation are transmitted via multiple signaling pathways. Eventually, the actin accumulation completes the membrane plug. The dynamics of actin assembly and subsequent disassembly are finely regulated by numerous ARPs, each recruited at specific times. In the final step, the repaired membranes may not precisely match the original unwounded membrane. Ultimately, these membranes are shed as cells migrate away, completing the wound repair process.

## 10. Conclusions and Perspective

The molecular mechanism underlying wound repair is intricate, and presently, no universal mechanism exists across different organisms. It is plausible that diverse mechanisms have evolved to ensure the survival of cells. Moving forward, it is crucial to approach research from the following perspectives: (1) Investigation of the membrane source for the membrane plug: Exploration into the origin of the membrane forming the membrane plug is warranted. If the membrane is generated de novo, examination of the mechanisms involved is necessary. (2) Exploration of force requirements for wound closure: Identification of the forces involved in wound closure and determination of the types of forces contributing to the process are necessary. Insights into these dynamics can be gained through direct measurements of force and tension around the wound pore. (3) Conducting ultrastructural observations: Essential observations of the wound pore closing process should be carried out using electron microscopy. Given the high repair speed, rapid freezing preparation is essential for accurate analysis. (4) Comprehensive identification of players: Utilization of genetic tools to comprehensively identify all factors related to wound repair is essential. Clarification of their spatiotemporal distribution is crucial for gaining a holistic understanding. (5) Conducting reconstitution experiments: Wound repair experiments using artificial membranes, including essential components, should be performed in vitro. (6) Conducting research on cell membrane damage in a wider variety of organisms. This will enable us to understand how organisms have acquired mechanisms for cell membrane damage. (7) Assessment of contributions to applied fields: Evaluation of the potential applications of research in this field, particularly in generating therapeutic medicine for the rapid recovery of diseases characterized by impaired wound repair, is necessary. Exogenous delivery of recombinant repair proteins such as MG53, annexins, or synthetic molecules has been shown to significantly enhance membrane repair in vivo and has proven effective for the treatment of muscular and neuronal damage [17,228,229,230,231]. Additionally, the exploration of implications for developing agricultural fungicides is warranted.

## Figures and Tables

**Figure 1 cells-13-00341-f001:**
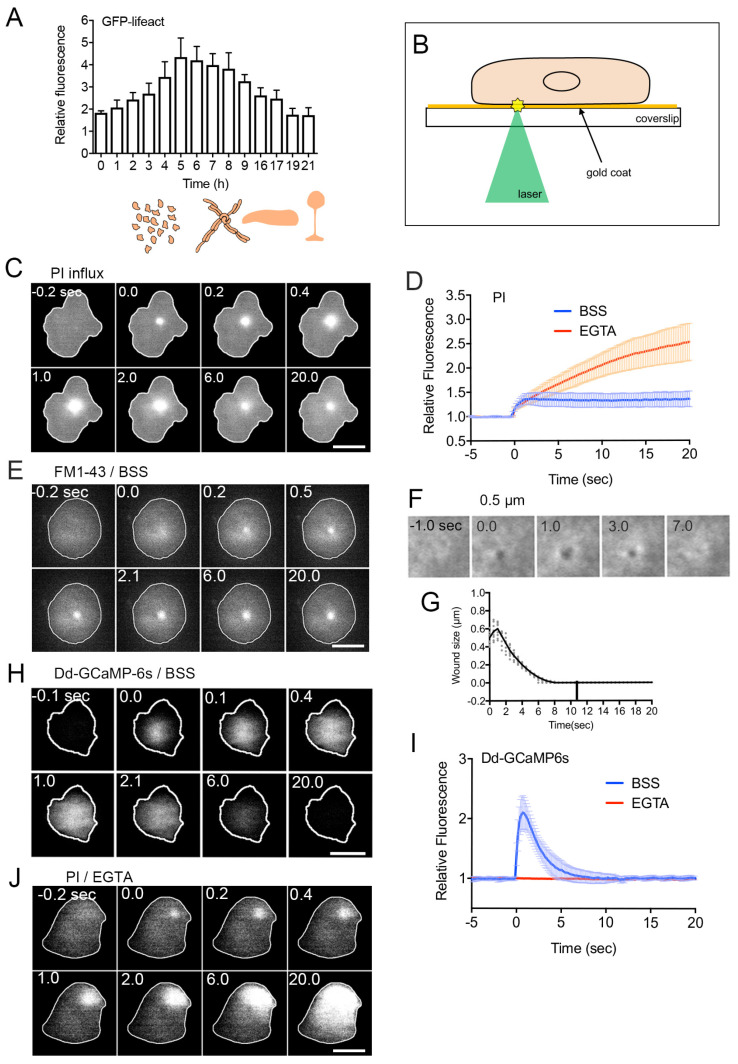
Wound repair of the cell membrane. (**A**) Relative amplitudes of actin accumulation at the wound site over time following starvation (0 h). As illustrated in the lower drawings, upon initiation of vegetative cell starvation, individual cells aggregate, forming streams direct toward the aggregation center. This process leads to the creation of a multicellular structure, culminating in the development of a fruiting body. Importantly, wound repair is observed at every stage of the lifecycle in *Dictyostelium discoideum*. (**B**) Schematic representation of the enhanced laserporation with gold coating. The wound diameter is usually set at 0.5 μm for *Dictyostelium* cells. (**C**) A representative sequence of fluorescence images capturing PI influx after laserporation. (**D**) Temporal profiles of PI influx in the presence (BSS, control) and absence (EGTA) of external Ca^2+^. The wound laser beam was applied at 0 sec. (**E**) A typical sequence of fluorescence images illustrating FM dye influx after laserporation. (**F**) Laserporation of a cell expressing GFP-cAR1 resulted in the appearance of a black spot on the cell membrane. The black spot transiently expanded, then contracted, and finally closed. (**G**) The time course of the black spot diameter. (**H**) A sequence of fluorescence images featuring a cell expressing GCAMP6s after laserporation. (**I**) Temporal profiles of GCAMP6s fluorescence intensities in the presence (BSS) and absence (EGTA) of external Ca^2+^. (**J**) A typical sequence of fluorescence images illustrating PI influx after laserporation in the absence (EGTA) of external Ca^2+^. Scale bars, 10 µm. Figures are posted from [48,51,67,68] with proper permission.

**Figure 2 cells-13-00341-f002:**
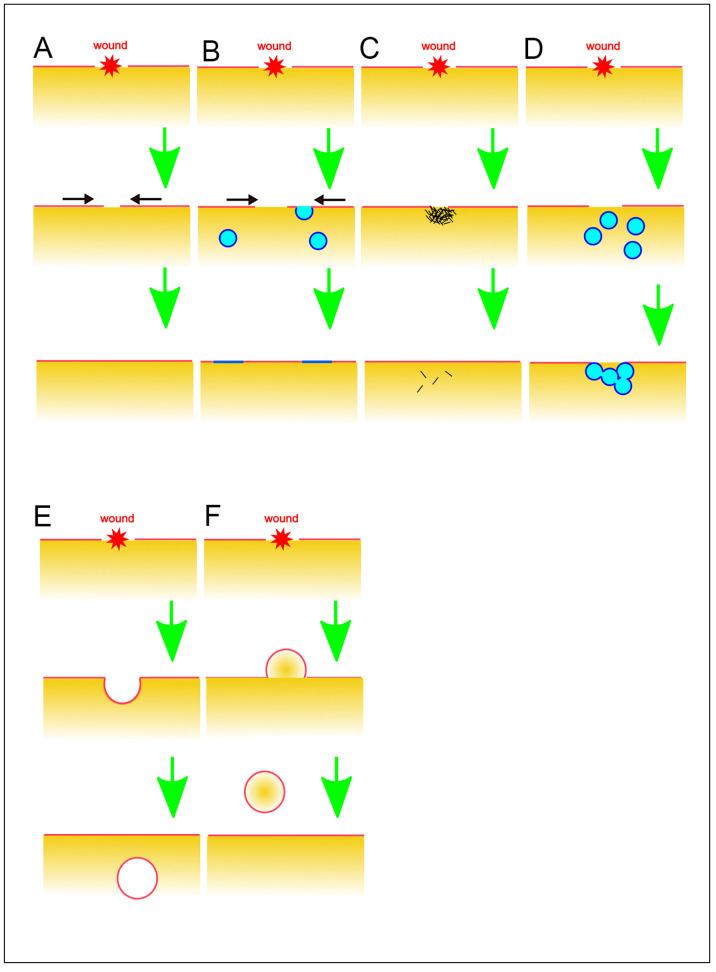
Various models for wound repair mechanisms. (**A**) Spontaneous self-sealing. (**B**) Self-sealing by regulation of surface tension. Black arrows indicate the direction of the membrane flow. (**C**) Sealing by protein aggregation. (**D**) Sealing by membrane patch. (**E**) Endocytosis of damaged membrane. (**F**) Vesicle budding and shedding to the outside. These illustrations are simplified for a better understanding of basic concepts.

**Figure 3 cells-13-00341-f003:**
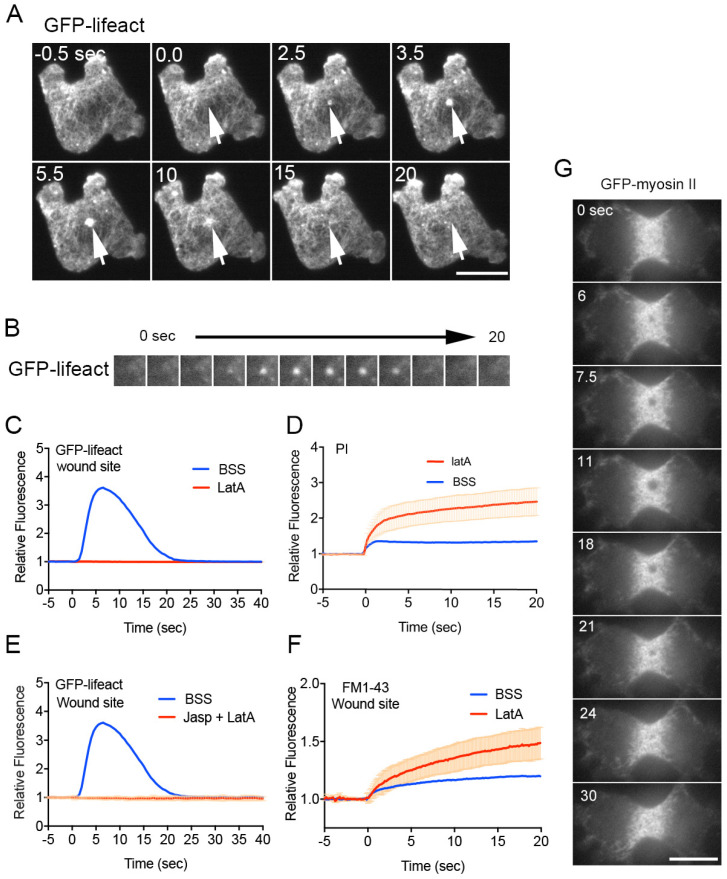
Role of actin in wound repair. (**A**) Representative sequence of fluorescence images featuring a cell expressing GFP-lifeact upon wounding. Arrows indicate the wound site. (**B**) Sequence of fluorescence images at the wound sites of cells expressing GFP-lifeact. (**C**) Temporal profiles of relative fluorescence intensity of GFP-lifeact at the wound site in the presence (LatA) and absence (BSS) of latrunculin A. (**D**) Temporal profiles of PI influx in the cytosol in the presence (LatA) and absence (BSS) of latrunculin A. (**E**) Temporal profiles of GFP-lifeact fluorescence intensities at the wound site in the presence and absence of jasplakinolide and latrunculin A (Jasp + LatA). (**F**) Temporal profiles of FM fluorescence intensities at the wound site in the presence (LatA) and absence (BSS) of latrunculin A. (**G**) Representative sequence of fluorescence images of a dividing cell expressing GFP-myosin II upon wounding. Scale bars, 10 µm. Figures are posted from [48,51,67,68] with proper permission.

**Figure 4 cells-13-00341-f004:**
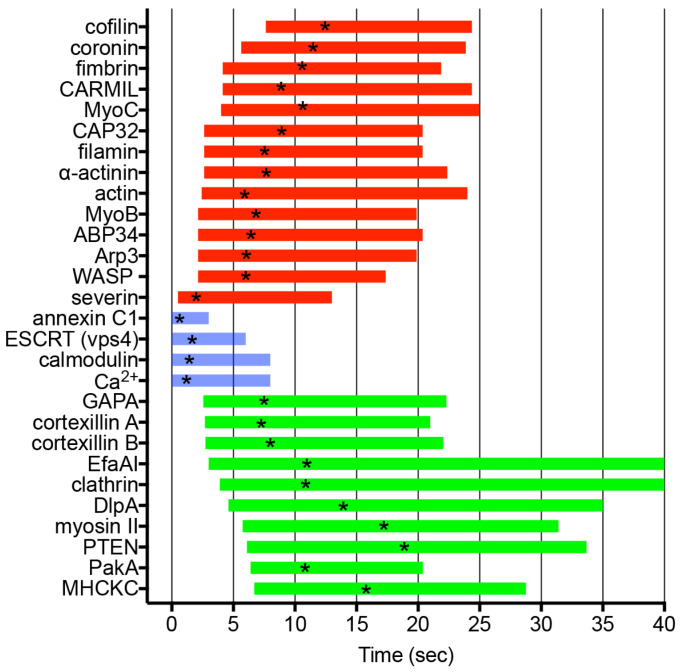
Dynamics of ARPs and signal-related proteins. The graph illustrates the duration of appearance (red) and disappearance (green) of individual ARPs and signal-related proteins, including actin at wound sites. Additionally, the durations of Ca^2+^ influx, calmodulin dynamics, ESCRT component vps4 dynamics, and annexin C1 dynamics are depicted (blue). Asterisks in the duration bars indicate peak times.

**Figure 5 cells-13-00341-f005:**
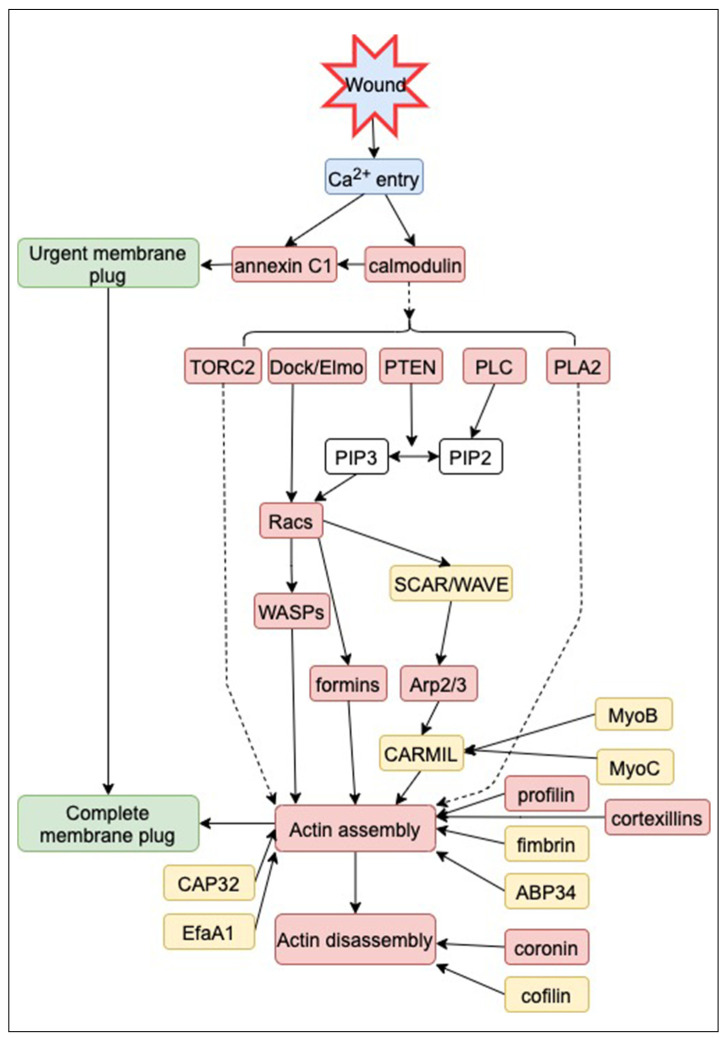
Signaling pathways for wound repair in *Dictyostelium*. ARPs and signal-related proteins contributing to wound repair, identified through null mutants or pharmacological inhibitors, are highlighted in red. Certain proteins (yellow) exhibit accumulation or disappearance at the wound site, though their specific contributions are yet to be fully elucidated. Data are based on [48,51,67,68].

**Figure 6 cells-13-00341-f006:**
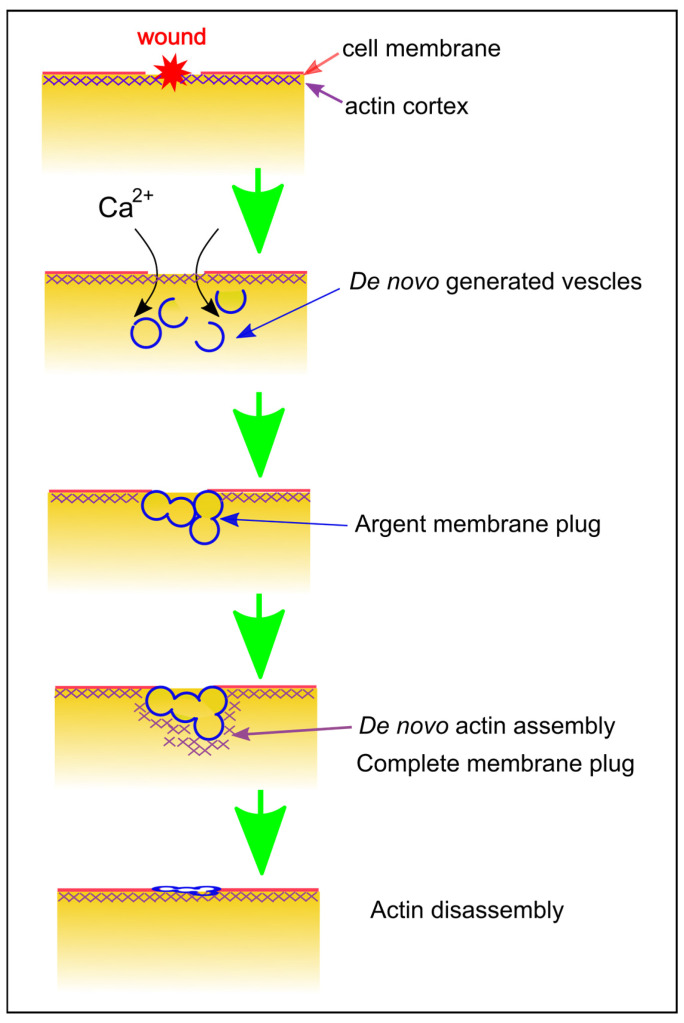
Summary of Wound Repair in *Dictyostelium* Cells. This schematic diagram illustrates the wound repair mechanism in *Dictyostelium* cells. Upon wounding, Ca^2+^ enters through the wound pore, initiating the de novo generation of vesicles and the mutual fusion of vesicle–vesicle and vesicle–cell membrane, forming an immediate membrane plug. Actin accumulates to finalize the plug, relying on Ca^2+^ and calmodulin. Following the disassembly of the actin structure, the remaining damaged membrane is shed as the cell undergoes migration.

## Data Availability

All relevant data are available from the authors on reasonable request.

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
