# Peer review of "Wound Repair of the Cell Membrane: Lessons from Dictyostelium Cells"

_cells, 2024, doi:10.3390/cells13040341_

Round 1

Reviewer 1 Report

Comments and Suggestions for Authors

This is a nice and comprehensive review on wound repair of cell membranes by Yumura, a leader in the field. The review thoroughly covers a wide variety of topics, including induction and detection of wound repair, progress of wound repair, as well as the various factors contributing to the repair of cell membrane wounds. This reviewer also appreciates the inclusion of the spontaneous self-sealing of model bilayers. I see no issue with this review and recommend acceptance of the manuscript.    

Author Response

I greatly appreciate your encouraging comments.

Reviewer 2 Report

Comments and Suggestions for Authors

The article of Yumura deals with the many mechanisms underlying the repair of cell membrane lesions. He discusses many methods of inducing membrane damage and many hypothesised mechanisms to repair it. In particular, he discusses the Dictyostelium-based cell model in detail. The article is long and would benefit from a reduction especially in the first part, the methods of injury induction, and a deepening in the perspectives part.  In line 271-272 it is not clear what the numbers after # are, and reference 7 needs to be corrected.

Author Response

First, I greatly appreciate your helpful comments.

C1: The article is long and would benefit from a reduction especially in the first part, the methods of injury induction, and a deepening in the perspectives part.

A1: I think that the methods of injury induction are helpful for general readers outside of this field. In the perspective part, I added some information of translation into clinical practice.

C2: In line 271-272 it is not clear what the numbers after # are, and reference 7 needs to be corrected.

A2: The mistakes of references were fixed.

Reviewer 3 Report

Comments and Suggestions for Authors

The Review "Wound Repair of the Cell Membrane: Lessons from Dictyoste- lium Cells" by Shigehiko Yumura summarizes the recent advancements in cellular wound repair, with a specific focus on the wound response and repair process in Dictyostelium cells. The Review is very interesting, original and well structured. However, it could be improved as follows:

A cartoon could clarify paragraph 2.

Paragraphs 6, 7and 8 could be summarized in a table.

Make sure that the full name of the genes and proteins, e.g. WASP etc., is included in the text for the first time.

Please include abbreviations list.

Please deepen the role of the Wiskott–Aldrich Syndrome protein (WASp) in the analyzed model considering its specifity for the haematopoietic system.

Please specify acronyms in the figures, especially in fig.5.

Are there pharmaceutics or natural molecules able to modulatete the wound repair mechanisms in the model analyzed?  Please deepen this aspect.

The author could summarize the message of the review in a graphical abstract.

Comments on the Quality of English Language

Minor editing of English language required

Author Response

First, I greatly appreciate your helpful comments.

C1: A cartoon could clarify paragraph 2.

A1: An explanatory cartoon of Dictyostelium lifecycle appears in the lower column of Figure 1A.

C2: Paragraphs 6, 7and 8 could be summarized in a table.

A2: Since the information of players involved in wound repair found in individual organisms are very fragmentated, I feel it is difficult to summarize them in a table.

C3: Make sure that the full name of the genes and proteins, e.g. WASP etc., is

included in the text for the first time.

A3: Mistakes are fixed.

C4: Please include abbreviations list.

A4: An abbreviation list is added as a supplementary file.

C5: Please deepen the role of the Wiskott–Aldrich Syndrome protein (WASp) in the analyzed model considering its specifity for the hematopoietic system. Please specify acronyms in the figures, especially in fig.5.

A5: According to the reviewer’s suggestion, some explanations of WASP are added. To specify acronyms, an abbreviation list is added as a supplementary file.

C6: Are there pharmaceutics or natural molecules able to modulate the wound repair mechanisms in the model analyzed? Please deepen this aspect.

A6: I mentioned effects of inhibitors. For example, an inhibitor of calmodulin inhibits the wound repair in Dictyostelium cells.  

C7: The author could summarize the message of the review in a graphical abstract.

A7: Thanks to the reviewer’s comment, a graphical abstract is added.

Reviewer 4 Report

Comments and Suggestions for Authors

The authors describe their work correctly. It is requested to improve the discussion paragraph by including how this information can be translated into clinical practice.

Author Response

First, I greatly appreciate your helpful comments.

C1: It is requested to improve the discussion paragraph by including how this information can be translated into clinical practice.

A1: In Conclusion and Perspective section, some information of translation into clinical practice by citing references is added as follows. 

Exogenous delivery of recombinant repair proteins such as MG53, annexins, or synthetic molecules has been shown to significantly enhance membrane repair in vivo and has proven effective for treatment of muscular and neuronal damages [17,228–231].

Round 2

Reviewer 3 Report

Comments and Suggestions for Authors

The authors significantly improved the manuscript. 

However, the required tables coud be increased the readibility of the review.

Comments on the Quality of English Language

Minor editing of English language is required